# Emerging Molecular Targets for Anti-Epileptogenic and Epilepsy Modifying Drugs

**DOI:** 10.3390/ijms24032928

**Published:** 2023-02-02

**Authors:** Katarzyna Łukasiuk, Władysław Lasoń

**Affiliations:** 1The Nencki Institute of Experimental Biology, Polish Academy of Sciences, 3 Pasteur St., 02-093 Warsaw, Poland; 2Maj Institute of Pharmacology, Polish Academy of Sciences, Smętna 12, 31-343 Kraków, Poland

**Keywords:** epilepsy, antiseizure medications, epileptogenesis, molecular mechanisms

## Abstract

The pharmacological treatment of epilepsy is purely symptomatic. Despite many decades of intensive research, causal treatment of this common neurologic disorder is still unavailable. Nevertheless, it is expected that advances in modern neuroscience and molecular biology tools, as well as improved animal models may accelerate designing antiepileptogenic and epilepsy-modifying drugs. Epileptogenesis triggers a vast array of genomic, epigenomic and transcriptomic changes, which ultimately lead to morphological and functional transformation of specific neuronal circuits resulting in the occurrence of spontaneous convulsive or nonconvulsive seizures. Recent decades unraveled molecular processes and biochemical signaling pathways involved in the proepileptic transformation of brain circuits including oxidative stress, apoptosis, neuroinflammatory and neurotrophic factors. The “omics” data derived from both human and animal epileptic tissues, as well as electrophysiological, imaging and neurochemical analysis identified a plethora of possible molecular targets for drugs, which could interfere with various stages of epileptogenetic cascade, including inflammatory processes and neuroplastic changes. In this narrative review, we briefly present contemporary views on the neurobiological background of epileptogenesis and discuss the advantages and disadvantages of some more promising molecular targets for antiepileptogenic pharmacotherapy.

## 1. Introduction

Epilepsy is a chronic neurological disease, which affects around 65 million people of all ages worldwide [1]. According to the official International League Against Epilepsy (ILAE, 2017) definition, “Epilepsy is characterized by repeated spontaneous bursts of neuronal hyperactivity and high synchronization in the brain”. The new ILAE classification of seizures is based on the site of seizure initiation in the brain (onset) comprising three major groups: seizures with focal onset, seizures with generalized onset and seizures with unknown onset. However, the previous nomenclature of seizures still exists and is used interchangeably, especially in preclinical research, e.g., temporal lobe epilepsy, absence epilepsy, or grand mal seizures.

Most epilepsies with infantile onset have a genetic origin and a plethora of monogenic forms have been described. The genes most frequently associated with epilepsy are those coding for voltage-dependent sodium, potassium and calcium ion channels [2,3]. Other epilepsy-associated gene variants are implicated in the disturbance of neurotransmitter release machinery (mutations of pre and postsynaptic receptors and transporters) and disorders in early brain development (proliferation, migration, dendritogenesis and synaptogenesis) [2,3,4]. Classical genetic analysis is often a complex, long and costly process. The next generation sequencing (NGS)-based tools allow genetic analyses to be performed faster, less expensively and at much higher resolution [4]. Although an imbalance between excitatory glutamatergic and inhibitory GABAergic transmission is thought to play the pivotal role in the mechanism of seizures, disturbances in other neurotransmitter systems can also contribute to the pathogenesis of epilepsy [5]. Currently, the treatment of epilepsy is purely symptomatic and aimed at seizure suppression. As reviewed by Sills and Rogawski, antiseizure medications (ASMs, formerly described as antiepileptic drugs) prevent and suppress seizures via several mechanisms, including blockade of voltage gated sodium and calcium channels, potentiation of GABAergic inhibition and suppressing excessive excitatory amino acid transmission [6] (Table 1). The seizure threshold appears to be closely associated with the magnitude of persistent Na^+^ current and suppression of this current may contribute to the mechanisms of antiepileptic therapies. As pointed out by Sills and Rogawski (2020), although the persistent current comprises only a small percentage of total Na^+^ conductance during depolarization, prolonged sodium channel late openings can contribute to the persistent depolarization that is reminiscent of the paroxysmal depolarizing shift, which reflects epileptiform activity. Recent studies have shed a new light on the mechanism of action of classical and newer ASMs. Thus, it was found that levetiracetam and breviracetam not only bind to SV2A proteins but may possess a multiple ionic mechanism in electrically excitable cells acting on M-type K^+^ current, large-conductance Ca^2+^-activated K^+^ channel and voltage-gated Na^+^ current [7]. Cannabidiol acts on G-protein coupled receptor GPR55, transient receptor potential cation channel TRPV1, voltage-gated sodium channels and equilibrative nucleoside transporter ENT1 as described by Sills and Rogawski (2020) [6]. The well-known regulatory actions of carbamazepine on ionic currents in electrically excitable cells has been recently reappraised by Wu et al. (2022) [8]. This drug increases fast-inactivated voltage-gated Na^+^ channels, modifies ionic currents (e.g., INa and erg-mediated K^+^ current [IK(erg)]). As far as rufinamide is concerned, the recent study by Lai et al. (2022) indicates that this drug stimulates Ca^2+^-activated K^+^ currents, while inhibiting voltage-gated Na^+^ currents. [9]. The oldest synthetic antiepileptic drug—phenobarbital—is a well-recognized positive allosteric modulator of GABA A receptors, especially on the δ-subunit-containing extrasynaptic GABA A receptors that mediate tonic inhibition and is an antagonist of AMPA receptors [6]. However, recent data indicated that independently of GABA A receptor allosteric modulation, phenobarbital could directly perturb the magnitude and gating of different plasmalemmal ionic currents including INa, IK(erg), IK(M) and IK(DR) [10].

There are also alternative treatments available, which include the ketogenic diet [11], vagal nerve stimulation [6] and surgical resection of the epileptic foci [12].

Despite the introduction of numerous new drugs on the market, around 30% of patients still suffer from drug-resistant epilepsy. The current understanding of the molecular, genetic and neuronal mechanisms of drug resistance in epilepsy and possible therapeutic options to overcome this problem have been extensively reviewed by Loscher et al., 2020) [13]. This percentage has not changed since the introduction of bromide as an antiepileptic drug at the end of the 19th century [14].

Many factors limit success in developing new methods of epilepsy pharmacotherapy. The majority of marketed ASMs have been discovered in animal screening tests, some by modification of chemical structures of an already known drug, e.g., brivaracetam, some of them were found serendipitously, e.g., valproate and only few of them were rationally designed based on their neurochemical mechanism of action e.g., vigabatrine and pregabalin [6]. Nevertheless, the progress in multiomics methods, as well as new strategies used for drug discovery including high-content screening, fragment-based drug discovery and virtual screening might accelerate identification of new molecules with favorable antiseizure activity. The development of therapies increasing the proportion of effectively treated patients may require the innovative selection of new targets for seizure suppression. Another approach to improve patient prognosis would be disease modification or even prevention. In this narrative review, we aim at summarizing such approaches and we indicate possible new directions in searching for new therapy targets.

For decades, targeting epileptogenesis has been a promising research direction aimed at preventing epilepsy. Epileptogenesis is defined as long-term, dynamic and progressive alterations in excitability and transformation of neuronal circuits, which ultimately lead to recurrent episodes of spontaneous (unprovoked) seizures [15]. In genetic epilepsies, epileptogenesis is regulated by neurodevelopmental changes in gene expression programming, which ultimately lead to the development of anomalous neuronal networks in adulthood [16]. Epileptogenesis can also be perceived as the process of formation of symptomatic epilepsy following an identified brain insult. In this case, it involves biochemical and morphological changes in the brain, which take place during the silent period after the insult but before the occurrence of the spontaneous window for potential antiepileptogenic therapies [17,18]. It has been elucidated that dynamic and progressive changes resulting in hyperexcitability and seizures reflect epileptogenesis. In the case of clinically relevant insults, such as stroke or traumatic brain injury, the latent period may last for weeks to years before the diagnosis of epilepsy, thereby providing a long time frame for the transformation of the neuronal network in the process of epileptogenesis, which comprises neurodegeneration, neurogenesis, gliosis, damage and sprouting of axons, dendritic plasticity, blood–brain barrier (BBB) damage, neuroinflammatory processes, reorganization of extracellular matrix and reorganization of the molecular structure of neurons (Table 2). These molecular mechanisms have been recently reviewed in several in-depth reviews [19,20,21]. Despite the accumulating knowledge, only a few molecular mechanisms have been targeted, to date, to prevent or modify the epilepsy development or phenotype, and no such therapy has as yet been developed. Moreover, it should be emphasized that an identified molecular target may not be clinically effective or development of the target-directed drugs may be hindered by many factors. Nevertheless, a better understanding of the neurochemical and molecular background of epileptogenesis seems a rational approach in designing antiepileptogenic strategies. What is of importance is that there is a relationship between epileptogenesis and the progression of epilepsy as brain damage-induced cellular and molecular changes may progress also after the occurrence of symptomatic seizures. Time dependency can be a crucial factor for epileptogenesis. As pointed out by Losher (2020), injury-induced epileptogenesis may have briefly opened a ‘therapeutic window’, whereas the delayed “secondary epileptogenesis” that affects the progression and refractoriness of seizures may be targeted by antiepileptogenic therapy even after onset of epilepsy [22]. Consequently, antiepileptogenic strategies can be directed not only at preventing seizures but also towards disease modification i.e., decreasing the frequency and time of seizures, changes in types of seizures and reversing drug resistance [20,22].

## 2. Neuronal Damage and Plasticity

Epileptogenesis has been most often studied in the hippocampus since it is the brain structure widely affected in epileptic patients, and particularly temporal lobe epilepsy was used as the animal model of this disease. A great number of morphological and neurochemical alterations related to epileptogenesis were reported, including selective neuronal loss, axonal and dendritic re-organization, dispersion of granule cells, changes in expression of ion channels and neurotransmitter receptors and astrogliosis. 

Regarding temporal lobe epilepsy, it has been hypothesized that the epileptogenic insult is followed by mossy fiber sprouting into the inner molecular layer and synapse with the proximal dendrites of neighboring granule cells, thereby establishing an aberrant positive-feedback excitatory circuit that generates seizure activity within the hippocampal formation [47]. However, there are controversies regarding the role of the reactive synaptogenesis and the existence of pre-epileptic “latent period” between brain injury and clinical epilepsy.

Based on animal models of temporal lobe epilepsy, the well-grounded “dormant basket cell” hypothesis assumes a selective loss of susceptible interneurons in the hilar region of the dentate gyrus [48]. Mossy fibers innervate both granule cells as well as the highly branched GABA interneurons [49]. As pointed out by Bumanglag and Sloviter (2018), the extensive hilar neuron loss and entorhinal cortical injury, which follow status epilepticus induced by perforant pathway stimulation, produce hyperexcitable granule cells, which immediately generate spontaneous epileptiform discharges and lead to focal or generalized behavioral seizures [33]. As they mentioned, the less extensive cell loss leads to an extended period during which initially subclinical focal seizures gradually increase in duration to produce the first clinical seizure. Thus, it is suggested that the “latent period” is rather a state of “epileptic maturation”; therefore, a promising antiepileptogenic strategy would be to interrupt this process [34]. Nevertheless, both in the “latent period of epileptogenesis” or during “epileptic maturation” it is important to fully understand their neuronal and biochemical background and to identify the most sensitive and appropriate molecular targets for halting the development of clinical seizures. Recently, Lybrand et al. (2021) identified a critical period for aberrant neurogenesis where sustained calcium transients are associated with abnormal maturation of adult-born granule cells promoting spontaneous seizures in a mouse model of epilepsy [28].

Since neuronal damage evoked by a mechanical, chemical or inflammatory insult initiates epileptogenesis, the agents which prevent various forms of cell death, e.g., necrosis, apoptosis, necroptosis, pyroptosic etc., might be considered the first line candidates for antiepileptogenic drugs. Of them, antioxidants, antagonists of ionotropic excitatory amino acid, NMDA, AMPA and kainate receptors and inhibitors of various cell death signaling pathways have been most extensively studied in animal models of epileptogenesis. It was found that glutamate receptor antagonists suppressed seizures but not epileptogenesis in animal models [23]. Substances which act through more than one molecular mechanism could be more efficient in halting the development of the pathological cascade, which leads to seizures. For example, 7,8-dihydroxyflavone, which possesses antioxidant and TrkB receptor agonist activity, was recently shown to exert an antiepileptogenic effect in the pilocarpine model of temporal lobe epilepsy [24].

The neurochemical bases of epileptogenesis were initially explored by evaluation of the changes in expression and functions of ion channels and receptors critically involved in controlling neuronal excitability in samples of brain tissues derived from epileptic patients and animal models of epilepsy. In this regard, studies in the pilocarpine model of temporal lobe epilepsy demonstrated a long-term decrease in the expression of voltage-dependent potassium channels (Kv3.4) and hyperpolarization-activated cyclic nucleotide-gated channels (HCN channels) in the hippocampus [50]. HCN channels are modulators of theta oscillations generated in the hippocampal and cortical neuronal networks and a recent study indicates that they are involved in the mechanism of antiepileptic effects of lamotrigine [51]. Although broad-spectrum inhibitors of HCN-mediated current (Ih), e.g., ivabradine, it has been reported that a HCN isoform-selective block can have a differential impact on seizure susceptibility [52]. Other investigators reported a decrease in the α1 subunit and an increase in the α4 GABA A receptor subunit in experimental epileptogenesis, which caused prevalence of the configuration of α4γ2 and α1γ2 [53]. 

Preclinical evidence suggests that the neurotrophin/Trk signaling pathways, in particular brain-derived neurotrophic factor (BDNF), can be a potential target for antiepileptogenic strategy. Neurotrophic factors modulate neurotransmitter release, expression of their receptors, enhance synaptogenesis and promote neuronal regeneration. They also play a role not only in physiological but also in pathological transformation of neuronal circuits, which could facilitate seizures. An enhanced BDNF expression in discrete subcellular domains of some brain structures seems to contribute to biochemical cascade of epileptogenesis. Electrophysiological studies showed that BDNF increased neuronal excitability and facilitated the development of experimental epileptogenesis [54]. It was reported that decreased BDNF signaling in transgenic mice with overexpression of truncated TrkB receptor reduced epileptogenesis [55,56]. On the contrary, some data indicate that BDNF can retard epileptogenesis through the stimulation of NPY expression or desensitization of TrkB [57]. An involvement of the BDNF signaling pathway in the mechanism of epileptogenesis likely depends on several factors, such as cellular characteristics of neuronal circuits and time- and structure-dependent expression of BDNF and its receptor [26]. Therefore, the utility of neurotrophin/Trk as a potential target for antiepileptogenic therapy needs to be verified in future studies. It has been hypothesized that distinct signaling pathways downstream of TrkB mediate neuroprotective effects and epileptogenesis. Indeed, evidence has been obtained that the TrkB-Shc-Akt signaling pathway is involved in TrkB-mediated neuroprotection, whereas TrkB-induced stimulation of phospholipase Cγ1 is required for epileptogenesis [25]. More recently, some aspiring strategies for targeting BDNF/TrkB signaling in order to prevent post-traumatic and poststatus epilepticus epilepsy have been reviewed and discussed by Lin et al. (2020) [27].

## 3. Astrogliosis

Seizure-induced hippocampal injury is accompanied by astrogliosis [58], and further studies have shown that interactions between glia cells and neurons appear to play a key role in the pathomechanism of both epileptogenesis and epilepsy [59]. Indeed, a growing body of evidence supports the notion about a pivotal role of astrocytes in the pathomechanism of epilepsy. Astrocytes synchronize the action potentials and maintain ionic balance through redistribution of potassium ions from regions with high neuronal activity to areas with low potassium ion concentrations and take up water through potassium (Kir4.1) and aquaporin (AQP4) channels. Astrocytes also remove excitatory neurotransmitters from the intrasynaptic space and participate in glucose metabolism. It has been shown that in epileptic tissue from patients with temporal lobe epilepsy and from animals after seizures, there are changes in expression, localizations and functioning of astroglial potassium channels, especially Kir 4.1, which affect the buffering ability of potassium ions. Moreover, in epileptic tissue, there are changes in the expression of aquaporin channels, accompanied by disturbances in glutamate transporters and glutamine synthase functioning [31,60,61]. Lee et al. (2012) found that mice devoid of the gene encoding AQP4 channel displayed more severe and more frequent kainate-induced seizures than wild type animals [30].

Astroglia and the TGFbeta signaling pathway can be also involved in epileptic changes connected with blood–brain barrier damage [39]. It has been suggested that a decrease in glutamine synthase activity, which metabolizes glutamate in astrocytes, can be responsible for early stages of epileptogenesis [62]. Several lines of evidence indicate that calcium-dependent glia transmission along with neuronal activity contribute equally to hypersynchronization of epileptogenic brain areas [63]. Glutamate released from neurons activates metabotropic glutamate receptors in astrocytes, which in turn increases the intracellular calcium ion concentration and release of glutamate from these cells resulting in neuronal hyperexcitability [63].

Besides glutamate, many other neuronal modulators are engaged in glia transmission, e.g., D-serine, ATP, adenosine and proinflammatory cytokine TNFalpha. Moreover, astrocytes synthesize GABA from putrescine and next the inhibitory amino acid is released via GAT-2 and GAT-3 GABA transporters and exerts a tonic inhibitory effect on excitable neuronal circuits. It has been found that glutamate uptake in astrocytes evokes GABA release by reversal of glial GAT-2 or GAT-3 GABA transporters. Of note, the magnitude of tonic inhibition provided by glial Glu/GABA exchange was more profound during seizure-like activity in rat entorhinal-hippocampal slices. It is not unlikely that targeting the molecular mechanism by which astrocytes transform glutamatergic excitation into tonic GABAergic inhibition mediated by high-affinity, slowly desensitizing, extrasynaptic GABA A receptors may have potential therapeutic implications in epilepsy [64]. 

It has been postulated that seizures originate from pathological high–frequency calcium waves generated by astrocytes [65]. This effect results in synchronization of neuronal discharges and amplifies the release of excitatory neurotransmitters, which further stimulate the generation of calcium waves in astrocyte syncytium. Accordingly, it has been proposed that the type of seizures could be determined by characteristics of astrocytic syncytium [65,66].

Among other glia transmitters, the deficit of which can decrease seizure threshold, adenosine deserves special attention. It was demonstrated that local disturbances of adenosine homeostasis in mice with focal seizures resulted in generalized seizures [67]. Adenosine, which is a potent endogenous anticonvulsant, is metabolized by adenosine kinase. It was suggested that astrogliosis and enhanced adenosine kinase expression could initiate epileptogenesis because decreased adenosine levels facilitated the generation of focal seizures. Transgenic mice with an overexpression of adenosine kinase are more prone to seizure occurrence and brain damage than wild-type mice, and adenosine kinase inhibitors diminish seizure generation. Moreover, intrahippocampal implants of stem cells devoid of adenosine kinase prevented epileptogenesis [32].

Among purinergic receptors, the ionotropic ATP-gated P2X7 receptor was suggested as a potential drug target for epilepsy treatment [68]. The ATP-gated P2X7 receptor is up-regulated in the hippocampus in epileptic patients and animal models of temporal lobe epilepsy. Moreover, an antagonist of this receptor showed a long-term inhibitory effect on seizures and gliosis in the hippocampus in experimental temporal lobe epilepsy suggesting that targeting the P2X7 receptor may have disease-modifying properties [69].

In order to find out whether astrogliosis is a cause or a consequence of epileptogenesis, Robel et al. (2015) used a mouse model of genetically induced, widespread chronic astrogliosis after conditional deletion of β1-integrin [29]. This study showed that astrogliosis was sufficient to induce epileptic seizures. Further experiments revealed that a shift in the relative expression of neuronal cation–chloride cotransporters NKCC1 and KCC2, similar to that observed in immature neurons during development, might contribute to astrogliosis-associated seizures.

## 4. mTOR Pathway

An important role of mTOR kinase in the pathomechanism of epilepsy has been postulated. This kinase is regulated in response to stressors, energetic status of the cell and growth factors. The mTOR kinase modulates growth, metabolism and proliferation of cells. Moreover, the mTOR signaling pathway modulates processes important for epileptogenesis, such as synaptic plasticity and ion channel protein expression [38]. It also regulates processes of cell survival, autophagy, apoptosis and immune functions. The selective inhibitors of mTOR include rapamycin (sirolimus), everolimus and temsirolimus, while caffeine and curcumin belong to less selective inhibitors of this enzyme.

As pointed out by Ostendorf and Wong (2015), the activity of the phosphoinositide 3-kinase (PI3K)/Akt/mammalian target of rapamycin (mTOR) (PI3K/Akt/mTOR) signaling pathway is enhanced in both human and animal epileptic brain tissues [35]. Furthermore, inhibition of this pathway activity leads to a reduction in seizure frequency in epileptic patients with mutations in TSC1 or TSC2 [70]. It has been postulated that the interaction of mTOR and inflammatory signaling pathways may play an important role in epileptogenesis, and consequently, simultaneous inhibition of both processes might be a potential antiepileptogenic strategy [36].

Recently, it was reported that microglial mTOR-deficient mice displayed increased neuronal loss and developed more severe spontaneous seizures after pilocarpine-induced status epilepticus, which suggests that microglial mTOR plays a protective role in mitigating neuronal loss and attenuating epileptogenesis in this model of temporal lobe epilepsy [37]. As the authors conclude, the observation that a deficiency of mTOR signaling in microglia increases seizure susceptibility contrasts the prevailing opinion that hyperactivation of mTOR is epileptogenic and might explain the inconsistent effect of rapamycin in animal models of epilepsy [37].

## 5. Neuroinflammation

Neuroinflammation has been detected in epileptogenic brain regions, and various inflammatory factors may be explored as potential targets for antiepileptogenic or antiepileptic therapies, including the IL-1 receptor-Toll-like receptor 4 axis, arachidonic acid-prostaglandin cascade and oxidative stress and transforming growth factor-β signaling associated with blood–brain barrier dysfunction [40]. Regarding the involvement of inflammation and oxidative stress in the pathomechanism of epilepsy, it was suggested that targeting molecular signaling pathways, such as the IL-1β-IL-1 receptor type 1 and TLR4, P2X7 receptors and transcriptional antioxidant factor Nrf2, did not prevent epilepsy development but might have clinically relevant disease-modifying effects [41]. Very recently, it has been reported that the connexin hemichannels in the glial cells can be a new target for putative antiepileptic drugs. Reactive glia cells release glutamate and other factors via the connexin hemichannels resulting in synapse modification, neuroinflammation and enhancement of seizure propagation [42].

## 6. Extracellular Matrix and Intercellular Communication

Since metalloproteinases (MMPs) and tissue inhibitors of MMPs are key players in the remodeling of the extracellular matrix, their involvement in epileptogenesis was also investigated. To this end, it was found that deletion and overexpression of the MMP9 gene reduced and increased susceptibility of mice to pentetrazole-induced kindling, respectively. Furthermore, an MMP9 deficit inhibited reactive synaptogenesis in the kainate model of temporal lobe epilepsy [44].

A possible involvement of integrins in epileptogenesis was also considered. Integrins are heterodimeric transmembrane receptors, which are engaged in processes of intercellular adhesion and adhesion of cells to extracellular matrix. These interactions are pivotal for neuroplastic changes, including the pathological ones, e.g., epileptogenesis [45]. Among other proteins, which are involved in the synaptic function, and dendritic spine development and remodeling, drebrins may also play a role in hippocampal synaptic reorganization during epileptogenesis. Immunohistochemical study showed that drebrin expression in the dentate gyrus of temporal lobe of epilepsy patients was associated negatively with seizure frequency and positively with verbal memory [46].

## 7. Epigenetic Mechanisms in Epileptogenesis

Epigenetics by definition means changes in gene function that occur without a change in the DNA sequence. Epigenetic regulation of gene transcription through changes in DNA methylation and histone modification may play a particular role in posttraumatic epilepsy, since numerous changes in DNA methylation and histone acetylation and methylation were detected in animal models of this disorder. The functional significance of epigenetics in the pathomechanism of epilepsy was supported by the observation that seizures occurred in HDAC4 knock-out mice [71]. Numerous epigenetic changes were found in animal models of epileptogenesis and epilepsy. To this end, in the pilocarpine model of temporal lobe epilepsy in rats, a decrease in H4 acetylation on the GluR2 promoter and increased acetylation on the BDNF P2 promoter was reported [72]. In the same model, an increase in phosphorylation of histone H3 was revealed [73]. In another model of temporal lobe epilepsy induced by kainate in mice, an enhanced acetylation of histone H4 0.5–6 h after seizures and increased histone H3 phosphorylation were detected [73,74]. Additionally, repeated electroconvulsive shock (a model of generalized tonic–clonic seizures) resulted in an increase in H4 acetylation on c-fos and BDNF promoters followed by a decrease in H4 acetylation after seizures [75].

Interestingly, the antiseizure drug valproate, which inhibits histone deacetylase, was shown to prevent aberrant neurogenesis and cognitive deficits in the kainate model of temporal lobe epilepsy [76]. Another study demonstrated that valproate protected neurons against seizure-induced damage and neurobehavioral deficits but did not prevent secondary epileptogenesis [77].

Temporal lobe epilepsy is frequently associated with hippocampal sclerosis, which worsens patient prognosis. Epigenetic analysis using whole-genome bisulfite sequencing (WGBS) showed 1171 hypermethylated and 2537 hypomethylated regions and found 632 differentially methylated genes (DMG) in the promoter region that participate in the modulation of epieptogenesis in temporal lobe epilepsy patients. Of note, the methylation profile was different in patients with hippocampal sclerosis as compared to patients without these pathological changes [78].

DNA hypo and hypermethylation was shown in three animal models of the temporal lobe epilepsy, namely pilocarpine-induced status epilepticus, amygdala stimulation-induced status epilepticus and traumatic brain injury [79]. This study showed differences in both methylation and gene expression between controls and epileptic animals. In particular, consistently increased methylation in gene bodies and hypomethylation in nongenic regions was observed; however, localization of specific methylation products differed between the models indicating distinct etiology-dependent DNA methylation patterns. This makes it difficult to think about designing a treatment strategy based on DNA methylation. Additionally, as pointed out by Berger et al. (2022), the development of antiepileptogenic drugs can be impaired by insufficient knowledge of molecular upstream mechanisms, such as DNA methylation of their potential targets as well as inflammatory processes and glial cell engagement in epileptogenesis [80]. These investigators reported neuron- and glia-specific changes in DNA methylation and gene expression, including epilepsy-related genes, such as HDAC11, SPP1, GAL, DRD1 and SV2C in the kainate model of early epileptogenesis in mice [81].

## 8. Transcriptomics

The introduction of large-scale methods for gene expression profiling, especially microarrays and RNA sequencing, allowed for the description of alterations in global gene expression patterns. Several datasets have been created over time at different times of epilepsy development, in different animal models and human disorders and from numerous brain structures. Analysis of this dataset individually and in meta-analyses allowed researchers to pinpoint the most prominent alterations in transcriptome and link them to biological functions, such as neuroinflammation, neurodegeneration, gliosis, proteolysis, neurogenesis and cell migration, extracellular matrix and other processes [20,82,83].

Additionally, miRNA, posttranscriptional regulators of gene expression, have been extensively studied in epileptic patients and animal models. Dysregulation of several relevant miRNAs has been described. These include miRNAs regulating genes involved in neurodegeneration, neuroinflammation, neurogenesis, signaling pathways, neuronal channels and receptors or neuronal plasticity [21,84,85,86,87,88,89,90]. 

More recently, another class of noncoding RNAs has been linked to epilepsy, that is lncRNA and circRNA. The role of these noncoding RNAs is much less understood than miRNAs. It is suggested that lncRNA regulate gene expression by a complex molecular mechanism, while circRNAs act as miRNA sponges [21]. Abnormal expression of lncRNA and circRNA has been observed both in patients and animal models of epilepsy [21,91,92,93,94]. There are data indicating that lncRNAs in epilepsy are involved in the regulation of synaptic plasticity, neuroinflammation and astro and microgliosis [21]. Much less is known on the role of circRNAs, but they have been shown to regulate miRNAs linked to epilepsy [21]. As noncoding RNAs are easily targetable, e.g., by oligonucleotide approach, there is a prospect of finding suitable drug targets in this class of molecules.

## 9. Proteomic Analysis in Studying Epileptogenesis

Proteomic analysis of epileptic tissues supports the hypothesis of a pivotal role of neuroplastic changes in epileptogenesis. To this end, Qian et al. (2022) performed proteomic analysis of the hippocampus derived from rats at 2 weeks after the onset of spontaneous seizures in the lithium–pilocarpine-induced temporal lobe epilepsy model. Out of total 4173 identified proteins, 27 were differentially expressed and these were related to the calmodulin-dependent protein kinase activity and were implicated in synaptic remodeling [95]. These data were confirmed by further study indicating the upregulation of CaMKII-α, CaMKII-β and GFAP proteins. In another study, a global protein expression analysis of the hippocampus from temporal lobe epilepsy patients and controls revealed differentially expressed proteins in the synaptic vesicle pathway, the prostaglandin synthesis and regulation pathway and endocannabinoid and retrograde modulation of synaptic transmission pathway. They found dysregulated expression of the excitatory amino acid transporter 1 (EAAT1), vesicular glutamate transporter 1 (VGLUT1) and up-regulation of the annexin family [96].

Regarding proteomic profile differentiation between mesial temporal lobe epilepsy with and without hippocampal sclerosis, Furukawa et al. (2020) conducted analyses of changes in protein expression level and protein oxidation status in the hippocampus or the neocortex in patients with mild and severe excitotoxic damages [97]. This study revealed decreased expression of d-3-phosphoglycerate dehydrogenase (an L-serine synthetic enzyme expressed only in astrocytes), increased expression of stathmin 1 (a neurite extension-related protein) and carbonylation of collapsin response mediator protein 2 (CRMP2). Pires et al. (2021) reported proteomic differences not only in the hippocampus but also in the cortex [98]. They found that brain tissue from epileptic patients showed significant differences in the expression of over one thousand proteins in comparison to control cases. Among proteins particularly altered in the epileptic tissues, there were proteins involved in protein synthesis, mitochondrial function, G-protein signaling and synaptic plasticity. Interestingly, the expression of G-protein subunit beta 1 was the most significantly decreased in epilepsy in all brain regions studied, pointing to a particular role of G-protein-coupled receptors in the pathomechanism of epilepsy. 

Using the multiomics approach, Canto et al. (2021) characterized the transcript and protein expression profile in the subregions of the hippocampal formation in the pilocarpine model of temporal lobe epilepsy in rats [99]. They found that most abnormalities in the transcript and protein levels occurred in the CA3 region and that abnormal regulation of N-methyl D-aspartate (NMDA) receptors, serotonin transmission, neuronal activity regulated by calcium/calmodulin-dependent protein kinase and leucine-rich repeat kinase 2 (LRRK2)/WNT signaling was mainly involved in the experimental epileptogenesis. The same researchers compared the proteomic signatures of the hippocampal lesion in three different animal models of temporal lobe epilepsy induced by pilocarpine, intracerebroventricular kainate and perforant pathway stimulation. The shotgun proteomics revealed that the pilocarpine model presented proteomic changes mainly related to neuronal excitatory imbalance, whereas the kainate and the perforant pathway stimulation resulted in changes related to the synaptic activity and metabolism/oxidative stress, respectively. However, a common feature of the proteomic analysis in these three models was that the alterations were related to inflammation and immune response, similar to those observed in the tissue derived from temporal lobe epilepsy patients. The authors concluded that only using a combination of the three models might replicate more closely the mechanisms of epilepsy in humans [100].

## 10. Attempts to Prevent Epileptogenesis

Various pharmacological strategies have been tried in order to prevent epileptogenesis, including seizure suppressing drugs. There is a general agreement that ASMs are not antiepileptogenic drugs and that they do not modify the course of epilepsy [101]. This notion is supported by outcomes of clinical trials, which showed that phenobarbital, phenytoin, valproate or carbamazepine did not prevent the development of post-traumatic epilepsy [102,103]. However, levetiracetam and ethosuximide were reported to prevent the development of seizures in animal models of genetic epilepsy [104,105], and brivaracetam showed antiepileptogenic effects in animal models of post-traumatic epilepsy. Furthermore, brivaracetam also shows disease-modifying properties suggesting a pivotal role of SV2A in the pathomechanism of epilepsy [106]. Recently, the antiepileptogenic effects of some antiepileptic drugs have been reviewed by Miziak et al. (2020) [107]. The authors stated that diazepam, gabapentin, pregabalin and, less clearly, valproate, protected experimental animals against acquired epilepsy and that the combination of levetiracetam + topiramate proved to be the most efficient in inhibiting status epilepticus-induced epileptogenesis [107]. Therefore, one can conclude that of new antiepileptic drugs, brivaracetam, gabapentin and pregabalin and a combination of levetiracetam + topiramate, at least in some models, show antiepileptogenic activity. Moreover, in a recent comprehensive review, Losher also included carbamazepine, phenytoin and valproate as antiseizure drugs with some antiepileptogenic properties, although no drug proved to be effective in preventing post-traumatic epilepsy [108].

Regarding anti-inflammatory and immunosuppressive drugs, it was found that tacrolimus (tacrolimus—immunophilin complex inhibits functional response to cytokines) had no effect in the model of temporal lobe epilepsy in rats [109]. Another immunosuppressant, rapamycin, is clinically used in the treatment of tuberous sclerosis, where it shows procognitive and anticonvulsant effects [110]. Regarding its effect on epileptogenesis, experimental studies have yielded mixed results. In particular, it was reported that rapamycin inhibited synaptic reorganization and recurrent excitation in neuronal loops of the dentate gyrus in a mouse model of temporal lobe epilepsy [111]. Yamawaki et al. (2015) observed that rapamycin inhibited excitatory synaptogenesis with proximal dendrites of dentate granule cells in the pilocarpine model of temporal lobe epilepsy in mice [112]. However, this compound failed to inhibit seizure generation in the model of temporal lobe epilepsy induced by electrical stimulation of the amygdala [113], and the authors suggested that the antiepileptogenic action of rapamycin might be limited to certain experimental models or experimental conditions. Other studies showed that rapamycin inhibited synaptic reorganization in the dentate gyrus but had no effect on the frequency of unprovoked seizures in the pilocarpine model of temporal lobe epilepsy in mice. Moreover, it did not protect hilar dentate gyrus cells against seizure-related damage in this model of TLE [114,115]. Regarding other immunossupressive mTOR inhibitors, it was found that repeated administration of everolimus did not prevent or ameliorate spontaneous recurrent seizures after kainic acid-induced status epilepticus in rats [116].

Among COX inhibitors, celecoxib decreased the frequency and duration of seizures and attenuated neuronal damage in the pilocarpine model of temporal lobe epilepsy [117]. Parecoxib did not affect frequency and duration of pilocarpine-induced seizures but attenuated their behavioral symptoms [118]. The COX-2 inhibitor SC58236 was without effect on seizure development in an electrical model of temporal lobe epilepsy [119]. Inhibition of leukocyte adhesion using monoclonal anti-integrin alpha4 antibody reduced the frequency of seizures, ameliorated blood–brain barrier damage and neuronal degeneration and improved behavioral parameters in a mouse model of temporal lobe epilepsy [120].

The antiepileptogenic potential of the anti-inflammatory drug minocycline has also been studied. Minocycline administered for 14 days after pilocarpine-induced status epilepticus in rats reduced neurodegeneration, microgliosis and the frequency, duration and severity of spontaneous recurrent seizures [43], although there are also contradictory reports [121].

Interestingly, some drugs which decrease the seizure threshold (proepileptic), such as the alpha 2 noradrenergic receptor antagonist atipamezol and rimonabant (an antagonist of CB1 cannabinoid receptor), inhibited epileptogenesis in animal models of status epilepticus and traumatic brain injury [122,123]. Out of the cannabinoids, repeated administration of cannabidiol showed both anticonvulsant and antiepileptogenic effects in genetic model of epilepsy (the Wistar Audiogenic Rat strain) by preventing neuronal hyperactivity in brain structures associated with tonic–clonic and limbic seizures [124].

There are also indications that some drugs currently on the market for other indications may become of use for antiepileptogenic treatment or epilepsy modification [22]. For example, it has been noted in clinical observational study that statins reduce the risk of epilepsy [125,126]. Another such compound, losartan, is an angiotensin II type 1 receptor (AT1) antagonist, a widely used drug for the treatment of hypertension and heart failure. It has been shown to prevent the development of delayed recurrent spontaneous seizures in rat models of vascular injury and in the kainic acid-induced epileptogenesis in rats [127,128].

Beyond pharmacotherapy, gene therapy may be a modern approach to the treatment of pharmacoresistant and genetic forms of epilepsy and possibly may also be useful to prevent epileptogenesis and modify epilepsy course (causal treatments). To this end, it has been demonstrated that overexpression of the endogenous anticonvulsant neuropeptide Y in the brain using viral vectors can suppress seizures in animal models of epilepsy [129,130]. Another gene frequently used for gene therapy, the *Kcna1* gene, encodes the voltage-gated potassium channel subunit Kv1.1. When introduced by adeno-associated virus (AAV), it was effective in decreasing the frequency and duration of seizures in animal models [130]. The influence of gene therapy on seizure generation has also been shown for GDNF, preprosomatostatin, GAD67 or galanin [130].

Recently, a new gene therapy approach has been proposed by Qiu et al. who constructed molecular closed-loop feedback system consisting of *Kcna1* under control of the Fos gene promoter, which is activated by neuronal activity [131]. This activity-dependent construct was introduced to the mouse hippocampus using adeno-associated virus (AAV) and caused suppression of pentylenetetrazole evoked seizures and decreased spontaneous seizure numbers in kainic acid-induced chronic epilepsy [131].

More daring, seemingly niche ideas also appear in the literature, for example, the intestinal microbiome was proposed as a therapy target. A significant difference in the composition of the gut microbiota among adult patients with drug-responsive and drug-resistant epilepsy has been observed indicating the role of the microbiome [132]. Interestingly, short treatment of drug-resistant epilepsy patients with ciprofloxacin, altering the composition of the gut microbiota, resulted in a reduction of seizure frequency [133]. Fecal microbiota transfer has been attempted in traumatic brain injury model in rats indicating that the gut microbiome may actively modulate the susceptibility to epilepsy [134]. As mentioned by Al-Beltagi and Saeed (2022), apart from fecal microbiota transplantation, a number of other gut manipulations, such as prebiotics, probiotics, synbiotics, postbiotics, vagus nerve stimulation and a ketogenic diet, were used successfully to manage epilepsy. It has been hypothesized that a low-carbohydrate high-fat ketogenic diet (KD) might prevent epileptogenesis through augmenting adenosine signaling in the brain. The investigators found that KD prevented disease progression in two mechanistically different models of epilepsy, and these therapeutic effects were associated with an increased adenosine level and a long-lasting decrease in DNA methylation [135]. It has been hypothesized that a high-fat low-carbohydrate ketogenic diet (KD) might prevent epileptogenesis through augmenting adenosine signaling in the brain. The investigators found that KD prevented disease progression in two mechanistically different models of epilepsy, and these therapeutic effects were associated with increased adenosine level and long-lasting decreased DNA methylation [136]. This study supports the notion on the importance of epigenetic mechanisms in potential antiepileptogenic therapies.

## 11. Conclusions

Despite the introduction of novel antiseizure medications in the last decades, there has been no significant improvement in seizure control in epileptic patients [101]. On the other hand, the tremendous progress in understanding the molecular and neuroanatomical background of epileptogenesis, as well as new drug screening strategies based on combinatorial chemistry, provide hope for developing antiepileptogenic and/or epilepsy-modifying drugs. Currently, the most promising molecular targets for such drugs are proteins crucially involved in the regulation of neuroplastic and neuroinflammatory processes. Furthermore, as very recently pointed out by Loscher and Klein (2022), multitarget and combinatorial drug therapies and synergistic combinations of repurposed drugs may be a more promising approach for preventing epilepsy [137]. In our opinion, when looking for antiepileptogenic drug targets, the attention should be paid especially to synaptic and extrasynaptic excitatory and inhibitory amino acid receptors, metabotropic glutamate receptors, adenosine receptor subtypes, neuropeptide receptors, proinflammatory cytokines and chemokine receptors. Furthermore, multitargeted drugs, which can act at distinct stages of epileptogenesis or rational combinations of single molecular target drugs that prevent pathological excitability and formations of seizure-maintaining neuronal circuits should be considered. Advances in molecular biochemistry, modern electrophysiology, neuroimaging, gene therapy, optogenetic tools and connectomics offer hope for “rewiring” the pathological seizure-generating or seizure-maintaining neuronal circuits to recover their physiological function. Furthermore, a translational approach, including molecular and functional analysis of human epileptic tissue and tissue derived from the most relevant animal models of epileptogenesis and epilepsy, might be helpful in designing antiepileptogenic and true antiepileptic drugs.

## Figures and Tables

**Table 1 ijms-24-02928-t001:** Currently available antiseizure medications and their mechanisms of action.

Name	Class of Drugs	Indications	Mechanism of Action
Phenobarbital	Barbiturate	All types of seizures, except absence seizures	A positive allosteric modulator of GABA A receptor, especially on δ-subunit containing extrasynaptic GABA A receptors that mediate tonic inhibition, an antagonist of AMPA receptors, affects ina, IK(erg), IK(M) and IK(DR) ionic currents
Primidone	Deoxybarbiturate	Focal-onset seizures, primary generalized seizures	Increases fast inactivation of voltage-gated Na^+^ channels
Phenytoin	Hydantoin	Focal-onset seizures, primary generalized seizures	Increases fast inactivation of voltage-gated Na^+^ channels
Ethosuximide	Succinimide	Absence seizures	Inhibits low voltage-gated calcium channels (T-type)
Carbamazepine	Iminostilbene	Focal-onset seizures, primary generalized seizures	Increases fast inactivation voltage-gated Na^+^ channels, modifes ionic currents (e.g., ina and erg-mediated K^+^ current [IK(erg)])
Oxcarbazepine	Iminostilbene	Focal-onset seizures, primary generalized seizures	Increases fast inactivation of voltage-gated Na^+^ channels
Eslicarbazepine	Iminostilbene	Focal-onset seizures	Increases fast inactivation voltage-gated Na^+^ channels
Clonazepam	Benzodiazepine	Focal-onset seizures, primary generalized seizures, status epilepticus	A positive allosteric modulator of GABA A receptor
Clobazam	Benzodiazepine	Focal-onset seizures, primary generalized seizures, Lennox–Gastaut syndrome, Infantile spasms (West syndrome), Dravet syndrome	A positive allosteric modulator at the GABA A receptor with some activity at sodium channels and voltage-sensitive calcium channels
Vigabatrin	GABA analogue	Focal-onset seizures, primary generalized seizures, infantile spasms (West syndrome)	Gabaergic inhibition, inhibition of excitatory neurotransmission, modulation of voltage- and receptor-gated calcium ion channels activity. An inhibitor of gamma-aminobutyric acid aminotransferase (GABA-AT)
Tiagabine	(R)-nipecotic acid	Focal-onset seizures	A GABA reuptake inhibitor blocking the GABA transporter 1 (GAT-1)
Gabapentin	Cyclic GABA analogue	Focal-onset seizures, mixed seizure disorders	A modulator of the presynaptic release machinery via α2δ subunit of calcium channels
Pregabalin	Cyclic GABA analogue	Focal-onset seizures	An inhibitor of α2δ subunit-containing voltage-dependent calcium channels
Cenobamate	Carbamate	Focal-onset seizures	Inhibits persistent sodium currents (inap)
Felbamate	Carbamate	Focal-onset seizures, Lennox–Gastaut syndrome, infantile spasms (West syndrome)	An allosteric modulator of GABA A receptors, an antagonist of NMDA receptors
Lamotrigine	Phenyltriazine	Focal-onset seizures, primary generalized seizures, Lennox–Gastaut syndrome, Infantile spasms (West syndrome)	Increases fast inactivation of voltage-gated Na^+^ channels, An HCN channel modulator,
Levetiracetam	Racetam	Focal-onset seizures, primary generalized seizures, Lennox–Gastaut syndrome, Dravet syndrome	Binds to the synaptic vesicle glycoprotein 2A (SV2A), inhibits presynaptic calcium channels
Brivaracetam	Racetam	Focal-onset seizures, primary generalized seizures	Binds to the SV2A protein, multiple ionic mechanism in electrically excitable cells including M-type K^+^ current, large-conductance Ca^2+^-activated K^+^ channel and voltage-gated Na^+^ current
Acetazolamide	Sulfonamide	Focal-onset seizures, primary generalized seizures, Lennox–Gastaut syndrome	Carbonic anhydrase inhibitor
Zonisamide	Sulfonamide	Focal-onset seizures, primary generalized seizures, Lennox–Gastaut syndrome, infantile spasms (West syndrome), Dravet syndrome	Increases fast inactivation of voltage-gated Na^+^ channels, modulates gabaergic and glutamatergic neurotransmission
Lacosamide	Functionalized amino acid	Focal-onset seizures, primary generalized seizures	Increases slow inactivation of voltage-gated Na^+^ channels
Valproate	Aliphatic carboxylic acid	Focal-onset seizures, primary generalized seizures, Lennox–Gastaut syndrome, infantile spasms (West syndrome), Dravet syndrome	Several mechanisms of action proposed including: inhibition of voltage-gated sodium channels, inhibition of GABA transaminase and succinate semialdehyde dehydrogenase, reduces release and/or effects of excitatory amino acids
Rufinamide	Triazole	Focal-onset seizures, primary generalized seizures, Lennox–Gastaut syndrome	Increases fast inactivation of voltage-gated Na^+^ channels, Stimulates Ca^2+^-Activated K^+^ currents while inhibiting Voltage-Gated Na^+^ currents
Stiripentol	Phenylpropanoid	Focal-onset seizures, primary generalized seizures, Lennox–Gastaut syndrome, Dravet syndrome	A positive allosteric modulator of GABA A receptor, interferes with GABA reuptake and metabolism
Perampanel	Bipyridine	Focal-onset seizures, primary generalized seizures	A selective non-competitive antagonist of ampa receptors
Topiramate	Sulfamate-substituted monosaccharide	Focal-onset seizures, primary generalized seizures, Lennox–Gastaut syndrome, Dravet syndrome	Several mechanisms of action proposed including: modulation of voltage-dependent sodium channels, potentiation of gabaergic inhibition, inhibition of excitatory neurotransmission, modulation of voltage- and receptor-gated calcium ion channels activity
Fenfluramine	Phenetylamine	Dravet syndrome, Lennox–Gastaut syndrome	A serotonergic 5-HT2 receptor agonist and σ1 receptor antagonist
Cannabidiol	Cannabinoid	Dravet syndrome, Lennox-Gastaut syndrome	Acts on G-protein coupled receptor GPR55, transient receptor potential cation channel TRPV1, voltage-gated sodium channels, and equilibrative nucleoside transporter ENT1
Everolimus	Macrocyclic lactone	Tuberous sclerosis complex-associated focal-onset seizures	Inhibitor of the mTORC1

**Table 2 ijms-24-02928-t002:** Possible molecular targets for antiepileptogenic therapy.

Stage of Epileptogenesis	Molecular Targets for Antiepileptogenic Therapy	Putative Antiepileptogenic Agents	References
Insult, excitotoxicity, oxidative stress, apoptosis/necrosis, neuronal damage	Glutamate receptors, reactive oxygen species, proapoptotic proteins, caspases, calpains	Glutamate receptor antagonists, antioxidants, antiapoptotic agents, protease inhibitors	Twele et al., 2015 [23]; Guarino et al., 2022 [24];
Neurogenesis	* BDNF/TrkB signaling pathway, TrkB-Shc-Akt signaling pathway	TrkB agonists, TrkB-Shc-Akt acivators	Guarino et al., 2022 [24]; Huang et al., 2019 [25]; Koyama and Ikegaya, 2005 [26]; Lin et al., 2020 [27]; Lybrand et al., 2021 [28]
Astrogliosis	Neuronal cation-chloride cotransporters NKCC1 and KCC2, aquaporin AQP4 channels, glutamate transporters, glutamine synthase	Modulators of astroglial potassium channels, especially Kir 4.1, adenosine kinase inhibitors, antagonists of ATP-gated P2X7 receptor	Robel et al., 2015 [29]; Lee et al., 2012 [30]; Steinhäuser et al., 2012 [31]; Boison, 2008 [32]
Axonal sprouting, dendritic plasticity	Neurotrophin/Trk signaling pathways, mTOR signaling pathway	mTOR inhibitors, e.g., rapamycin	Bumanglag and Sloviter, 2018 [33]; Sloviter and Bumanglag, 2013 [34]; Ostendorf and Wong, 2015 [35]; Hodges and Lugo, 2020 [36]; Zhao et al. 2020 [37]; Citraro et al., 2016 [38]
Blood–brain barrier damage	TGFbeta signalling pathway, gap junction proteins, potassium channels	Modulators of TGFβ signaling,	Heinemann et al., 2012 [39]
Neuroinflammation	The IL-1 receptor-Toll-like receptor 4 axis, COX-2, transforming growth factor-β signalling	Antagonists of proinflammatory cytokine receptors, COX-2 inhibitors, modulators of connexin hemichannels, minocycline,	Vezzani et al., 2019 [40]; Terrone et al., 2020 [41]; Guo, et al., 2022 [42]; Wang et al., 2015 [43]
Reorganization of extracellular matrix	Integins, drebrins	Tissue inhibitors of MMPs	Wilczynski, 2008 [44]; Wu and Reddy, 2012 [45]; Dombroski etal., 2020 [46]
Drug-resistant epilepsy	P-glycoprotein (P-gp)	Pgp inhibitors, e.g., tariquidar	Loscher et al., 2020 [13]

* BDNF shows both pro and antiepileptogenic effects. An engagement of the BDNF signaling pathway in the mechanism of epileptogenesis appears to depend on cellular characteristics of neuronal circuits and time- and brain region-dependent expression of BDNF and its receptor.

## Data Availability

Not applicable.

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
