# Peer review of "Emerging Molecular Targets for Anti-Epileptogenic and Epilepsy Modifying Drugs"

_ijms, 2023, doi:10.3390/ijms24032928_

Round 1
Reviewer 1 Report
In this review article, the authors present contemporary views on the neurobiological background of epileptogeneis and possible mechanisms of anti-epileptogenic compounds. In general, the manuscript has been well written. There are several comments which needs to be pointed out.
(1) In Table 1, the antiepileptic effect of brivaracetam should be also associated with its perturbations on ionic currents in excitable cells.
(2) In Table 1, cannabidiol effect on ionic currents needs to be stated as well.
(3) In Table 1, carbamazepine and rufinamide should be described as an inhibitor with increased inactivation of voltage-gated Na+ current, as reported recently.
(4) In Table 1, phenobarbital action could be not simply due to its modulation of GABA A receptor. Inhibitory effects of phenobarbital on different types of ionic currents also need to be pointed out.
(5) Overall, Table 1 needs to be well organized and additional information had better be incorporated in Table 1.
(6) A growing number of new papers, which is closely linked to regulatory actions of antiepileptic compounds or drugs on membrane ionic currents (e.g., voltage-gated Na+ current and late or persistent Na+ current), need to be quoted and stated.
(7) Of notice, seizure threshold is closely associated with the magnitude of persistent Na+ current. Please discuss this issue somewhere in the manuscript.
(8) The portion of page 6, lines 143-151, needs to be expanded to some extent. For example, some of antiepileptic agents could potentially suppress the magnitude of HCN-encoded current.
Author Response
In this review article, the authors present contemporary views on the neurobiological background of epileptogeneis and possible mechanisms of anti-epileptogenic compounds. In general, the manuscript has been well written. There are several comments which needs to be pointed out.
(1) In Table 1, the antiepileptic effect of brivaracetam should be also associated with its perturbations on ionic currents in excitable cells. Response: As suggested by the reviewer, we have now included in the Table 1 and in the text an additional information on brivaracetam mechanism of action. It reads: ” binds to SV2A proteins, multiple ionic mechanism in electrically excitable cells including M-type K+ current, large-conductance Ca2+-activated K+ channel and voltage-gated Na+ current. The relevant paper by Hung et al. (2021) has been cited.
(2) In Table 1, cannabidiol effect on ionic currents needs to be stated as well. Response: The Table 1 has been supplemented with the following information: “acts on G-protein coupled receptor GPR55, transient receptor potential cation channel TRPV1, voltage-gated sodium channels, and equilibrative nucleoside transporter ENT1” as described by Sills and Rogawski (2020).
(3) In Table 1, carbamazepine and rufinamide should be described as an inhibitor with increased inactivation of voltage-gated Na+ current, as reported recently. Response: Indeed, the regulatory actions of carbamazepine on ionic currents in electrically excitable cells has been recently reappraised (Wu et al., 2022). Accordingly the Table 1 has been supplemented with the following informations “ increases fast inactivation voltage-gated Na+ channels, modifies ionic currents (e.g., INa and erg-mediated K+ current [IK(erg)]). As far as rufinamide is concerned, the recent study by Lai et al.(2022) indicates that this drug stimulates Ca2+-activated K+ currents while inhibiting Voltage-Gated Na+ currents. This information has been added to Table 1 and on page 4.
(4) In Table 1, phenobarbital action could be not simply due to its modulation of GABA A receptor. Inhibitory effects of phenobarbital on different types of ionic currents also need to be pointed out. Response: Agreed. Recent data indicated that besides being GABA A receptor allosteric modulator, phenobarbital could directly perturb the magnitude and gating of different plasmalemmal ionic currents including INa, IK(erg), IK(M) and IK(DR) (Wu et al., 2022). This information has been added to Table 1 and to the text (page 4)
(5) Overall, Table 1 needs to be well organized and additional information had better be incorporated in Table 1. Response: The Table 1 has been re-organized by adding new information on antiepileptic drug mechanism of actions.
(6) A growing number of new papers, which is closely linked to regulatory actions of antiepileptic compounds or drugs on membrane ionic currents (e.g., voltage-gated Na+ current and late or persistent Na+ current), need to be quoted and stated. Response: Agreed. The manuscript has been enriched with the most recent and relevant publications: a) Wu, P.-M.; Cho, H.-Y.;Chiang, C.-W.; Chuang, T.-H.; Wu,S.-N.; Tu, Y.-F. Characterization in Inhibitory Effectiveness of Carbamazepine in Voltage-Gated Na+ and Erg-Mediated K+ Currents in a Mouse Neural Crest-Derived (Neuro-2a) Cell Line. Int. J. Mol. Sci. 2022, 23, 7892). b) Lai et al. Ca2+-Activated K+ Currents while inhibiting Voltage-Gated Na+ Currents. Int. J. Mol. Sci. 2022, 23(22), 13677 c) Po-Ming Wu , Pei-Chun Lai, Hsin-Yen Cho, Tzu-Hsien Chuang, Sheng-Nan Wu, Yi-Fang Tu. Effective Perturbations by Phenobarbital on INa, IK(erg), IK(M) and IK(DR) during Pulse Train Stimulation in Neuroblastoma Neuro-2a Cells Biomedicines. 2022 Aug 13;10(8):1968. d) Paulina Kazmierska-Grebowska, Marcin Siwiec, Joanna Ewa Sowa, Bartosz Caban, Tomasz Kowalczyk, Renata Bocian, M Bruce MacIver Lamotrigine Attenuates Neuronal Excitability, Depresses GABA Synaptic Inhibition, and Modulates Theta Rhythms in Rat Hippocampus. Int J Mol Sci. 2021 Dec 19;22(24):13604. e) Qays Kharouf, Paulo Pinares-Garcia, M Novella Romanelli, Christopher A Reid. Testing broad-spectrum and isoform-preferring HCN channel blockers for anticonvulsant properties in mice. Epilepsy Res. 2020 Dec;168:106484, Epub 2020 Oct 10). f) Wolfgang Löscher. Single-Target Versus Multi-Target Drugs Versus Combinations of Drugs With Multiple Targets: Preclinical and Clinical Evidence for the Treatment or Prevention of Epilepsy. Review. Front Pharmacol. 2021 Oct 27;12:730257) g) Te-Yu Hung, Sheng-Nan Wu, and Chin-Wei Huang. The Integrated Effects of Brivaracetam, a Selective Analog of Levetiracetam, on Ionic Currents and Neuronal Excitability. Biomedicines. 2021 Apr 1;9(4):369. h) Paulina Kazmierska-Grebowska, Marcin Siwiec, Joanna Ewa Sowa, Bartosz Caban, Tomasz Kowalczyk, Renata Bocian, M Bruce MacIver Lamotrigine Attenuates Neuronal Excitability, Depresses GABA Synaptic Inhibition, and Modulates Theta Rhythms in Rat Hippocampus. Int J Mol Sci. 2021 Dec 19;22(24):13604) i) Qays Kharouf, Paulo Pinares-Garcia, M Novella Romanelli, Christopher A Reid. Testing broad-spectrum and isoform-preferring HCN channel blockers for anticonvulsant properties in mice. Epilepsy Res. 2020 Dec;168:106484 Epub 2020 Oct 100 j) Mohammed Al-Beltagi, Nermin Kamal Saeed. Epilepsy and the gut: Perpetrator or victim? World J Gastrointest Pathophysiol. 2022 Sep 22;13(5):143-156. Review. k) Theresa A Lusardi , Kiran K Akula, Shayla Q Coffman, David N Ruskin, Susan A Masino, Detlev Boison. Ketogenic diet prevents epileptogenesis and disease progression in adult mice and rats. Neuropharmacology. 2015 Dec;99:500-9.
(7) Of notice, seizure threshold is closely associated with the magnitude of persistent Na+ current. Please discuss this issue somewhere in the manuscript. Response: We added the following sentences to the manuscript: Seizure threshold appears to be closely associated with the magnitude of persistent Na+ current and suppression of this current may contribute to the mechanisms of antiepileptic therapies. As pointed out by Sills and Rogawski (2020), although the persistent current comprises only a small percentage of total Na+ conductance during depolarization, prolonged sodium channel late openings can contribute to the persistent depolarization that is reminiscent of the paroxysmal depolarizing shift which reflects epileptiform activity.
(8) The portion of page 6, lines 143-151, needs to be expanded to some extent. For example, some of antiepileptic agents could potentially suppress the magnitude of HCN-encoded current. Response: After the sentence on the page 6 “In this regard, studies in the pilocarpine model of temporal lobe epilepsy demonstrated a long-term decrease in the expression of voltage-dependent potassium channels (Kv3.4) and hyperpolarization-activated cyclic nucleotide-gated channels (HCN channels) in the hippocampus [24]” we added the following fragment ”HCN channels are modulators of theta oscillations generated in the hippocampal and cortical neuronal networks and a recent study indicates that they are involved in the mechanism of antiepileptic effects of lamotrigine (Kazmierska-Grebowska et al., 2021). Although broad-spectrum inhibitors of HCN-mediated current (Ih), e.g., ivabradine, it has been reported that HCN isoform-selective block can have a differential impact on seizure susceptibility (Kharouf et al., 2020).”
Reviewer 2 Report
Review of Manuscript “Emerging molecular targets for anti-epileptogenic and epilepsy modifying drugs” submitted to International Journal of Molecular Sciences by Katarzyna Łukasiuk and Władysław Lasoń.
This was really an interesting review which presents contemporary views on the neurobiological background of epileptogenesis, as well as, some molecular targets for anti-epileptogenic pharmacotherapy. Overall, the paper is well-written and could be published the way it is.
My minor recommendations are listed below:
1. The idiom “pros and cons” used in the abstract can be replaced with something more scientific.
2. I miss a little bit more information about the anti-epileptogenic effects of the newer ASMs.
3. In addition, a little bit more information can be provided for the alternative methods applied for the treatment of epilepsy and their anti-epileptogenic properties.
Author Response
Reviewer 2
This was really an interesting review which presents contemporary views on the neurobiological background of epileptogenesis, as well as, some molecular targets for anti-epileptogenic pharmacotherapy. Overall, the paper is well-written and could be published the way it is.
My minor recommendations are listed below:
- The idiom “pros and cons” used in the abstract can be replaced with something more scientific. Response: As suggested by the reviewer, the idiom has been replaced with “advantages and disadvantages”.
- I miss a little bit more information about the anti-epileptogenic effects of the newer ASMs. Response: We now added the following information to the revised manuscript: “Therefore, one can conclude that of new antiepileptic drugs, brivaracetam, gabapentin and pregabalin and a combination of levetiracetam + topiramate at least in some models show antiepileptogenic activity. Moreover, in a recent comprehensive review Losher (2021) included also carbamazepine, phenytoin and valproate as antiseizure drugs with some antiepileptogenic properties, although no drug proved to be effective in preventing post-traumatic epilepsy.”
- In addition, a little bit more information can be provided for the alternative methods applied for the treatment of epilepsy and their anti-epileptogenic properties.
Response: We added the following information: “ As mentioned by Al-Beltagi and Saeed (2022), apart from fecal microbiota transplantation a number of other gut manipulations, such as prebiotics, probiotics, synbiotics, postbiotics, vagus nerve stimulation and a ketogenic diet were used successfully to manage epilepsy. It has been hypothesized that a low carbohydrate high fat ketogenic diet (KD) might prevent epileptogenesis through augmenting adenosine signaling in the brain. The investigators found that KD prevented disease progression in two mechanistically different models of epilepsy, and these therapeutic effects were associated with increased adenosine level and long-lasting decrease in DNA methylation [136]. It has been hypothesized that high fat low carbohydrate ketogenic diet (KD) might prevent epileptogenesis through augmenting adenosine signaling in the brain. The investigators found that KD prevented disease progression in two mechanistically different models of epilepsy, and these therapeutic effects were associated with increased adenosine level and long-lasting decreased DNA methylation [137]. This study supports the notion on the importance of epigenetic mechanisms in potential antiepileptogenic therapies. „
Reviewer 3 Report
There are some serious short-comings that need to be addressed: (1) absence of the work Of W Loscher (Pharmacol Rev 72:606–638, July 2020, and 10.1016/j.neuropharm.2019.04.011) that are aprticulalry pertinent to drug resistance; (2) The review on neurotransmitters by Akyuz et al (Life Sciences 265 (2021) 118826) and genetics by Fernández-Marmiesse (10.3389/fnins.2019.01135). These “omissions” significantly detract enthusiasm. Moreover, many of the mechanisms discussed are not unique to epilepsy and have had mixed results in other areas. The authors may wan to adapt their title as to be more focused.
1) The “tone” of the manuscript should be adjusted, many CNS disorders’ treatment is symptomatic! (for that matter essential hypertension has analogous approaches). Line 56-58 exemplifies this point. Identifying a target does not make it druggable, nor guarantees clinical success.
a. Line 83-85 a target may not be clinically effective, or its development be hindered by many factors.
2) Line 34 please give an example, Are the authors referind to Grand Mal etc ?
3) 50 % ethiological factors? Where does such high figure come from?
4) Table 1 consider re-arranging by mechanisms
5) Table 2 notable abscne of MDR overexpression by ASM ( Loscher)
6) Line 202-204. Even if one is familiar with the are it is not possible to guess what the authors mena. Are these phenomena linked ? if so how? If not ate least 2 independent sentences are needed.
7) Line 253-255 relationship between neuronal loss and microglial mTOR?
Author Response
Reviewer 3
There are some serious short-comings that need to be addressed: (1) absence of the work Of W Loscher (Pharmacol Rev 72:606–638, July 2020, and 10.1016/j.neuropharm.2019.04.011) that are aprticulalry pertinent to drug resistance; (2) The review on neurotransmitters by Akyuz et al (Life Sciences 265 (2021) 118826) and genetics by Fernández-Marmiesse (10.3389/fnins.2019.01135). These “omissions” significantly detract enthusiasm. Moreover, many of the mechanisms discussed are not unique to epilepsy and have had mixed results in other areas. The authors may wan to adapt their title as to be more focused. Response: : We thank the reviewer for noticing the lack of publications relevant to the subject of our work. This lack has been supplemented and the indicated publications have been cited in the Introduction. As far as the title is concerned, we suggest not changing it, as it reflects the content of the work well.
1) The “tone” of the manuscript should be adjusted, many CNS disorders’ treatment is symptomatic! (for that matter essential hypertension has analogous approaches). Line 56-58 exemplifies this point. Identifying a target does not make it druggable, nor guarantees clinical success. Response: We agree. Our statement was oversimplified. The sentence: „The limited success in developing new methods of pharmacotherapy may result from the fact that current drugs target a limited number of mechanisms, mostly related to receptors and channel function” has been changed to: „Many factors limit success in developing new methods of epilepsy pharmacotherapy. The majority of marketed ASMs have been discovered in animal screening tests, some by modification of chemical structures of already known drug, e.g. brivaracetam, some of them were found serendipitously, e.g. valproate and only few of them were rationally designed based on their neurochemical mechanism of action e.g. vigabatrine, pregabaline [6]. Nevertheless, the progress in multiomics methods, as well as new strategies used for drug discovery including high-content screening, fragment-based drug discovery and virtual screening might accelerate identification of new molecules with favorable anti-seizure activity.”
- Line 83-85 a target may not be clinically effective, or its development be hindered by many factors. Response: We agree. After the sentence „Despite the accumulating knowledge, only a few molecular mechanisms have been targeted to date to prevent or modify the epilepsy development or phenotype, and no such therapy has as yet been developed.” the whole fragmentof the text has been changed and now it reads: „Moreover, it should be emphasized that an identified molecular target may not be clinically effective, or development of the target-directed drugs may be hindered by many factors. Nevertheless, a better understanding of the neurochemical and molecular background of epileptogenesis seems a rational approach in designing antiepileptogenic strategies. What is of importance, there is a relationship between epileptogenesis and progression of epilepsy as brain damage-induced cellular and molecular changes may progress also after the occurrence of symptomatic seizures. Time dependency can be a crucial factor for epileptogenesis. As pointed out by Losher (2020), injury-induced epileptogenesis may have briefly opened a 'therapeutic window', whereas the delayed "secondary epileptogenesis" that affects the progression and refractoriness of seizures may be targeted by antiepileptogenic therapy even after onset of epilepsy [22]. Consequently, antiepileptogenic strategies can be directed not only at preventing seizures, but also towards disease modification i.e., decreasing the frequency and time of seizures, changes in types of seizures, and reversing drug resistance [20].”
2) Line 34 please give an example, Are the authors referind to Grand Mal etc ? „ However, the previous nomenclature of seizures still exists and is used interchangeably, especially in preclinical research”. Response; Yes, we refer to temporal lobe epilepsy, absence epilepsy or grand mal seizures.
3) 50 % ethiological factors? Where does such high figure come from?
Response: We appologize for this mistake. It has been removed in the revised manuscript. Many studies have assessed the etiologic factors of symptomatic epilepsies, but results are variable and they should be treated with a cautious. For example Symonds et al. reported: „ Aetiology was determined in 54% of children, and epilepsy syndrome was classified in 54%. Thirty-one per cent had an identified genetic cause for their epilepsy”( Symonds et al. Early childhood epilepsies: epidemiology, classification, aetiology, and socio-economic determinants. Brain. 2021 Oct 22;144(9):2879-2891. Brain. 2021 Oct 22;144(9):2879-2891)
4) Table 1 consider re-arranging by mechanisms. Response: The table has been re-arranged as suggested.
5) Table 2 notable abscne of MDR overexpression by ASM ( Loscher) Response: The table 2 has been supplemented with the suggested information.
6) Line 202-204. Even if one is familiar with the are it is not possible to guess what the authors mena. Are these phenomena linked ? if so how? If not ate least 2 independent sentences are needed. Response: We agree that the meaning of the sentences was not clear enough. Now it reads: Moreover, astrocytes synthesize GABA from putrescine, and next the inhibitory amino acid is released via GAT-2 and GAT-3 GABA transporters and exerts a tonic inhibitory effect on excitable neuronal circuits. It has been found that glutamate uptake in astrocytes evokes GABA release by reversal of glial GAT-2 or GAT-3 GABA transporters. Of note, the magnitude of tonic inhibition provided by glial Glu/GABA exchange was more profound during seizure-like activity in rat entorhinal-hippocampal slices. It is not unlikely that targeting the molecular mechanism by which astrocytes transform glutamatergic excitation into tonic GABAergic inhibition mediated by high-affinity, slowly desensitizing, extrasynaptic GABA A receptors may have potential therapeutic implications in epilepsy [51].
7) Line 253-255 relationship between neuronal loss and microglial mTOR? Response: The exact mechanism of relationship between neuronal loss and microglial mTOR has not been elucidated yet. After the sentence „Recently, it was reported that microglial mTOR-deficient mice displayed increased neuronal loss and developed more severe spontaneous seizures after pilocarpine-induced status epilepticus, which suggests that microglial mTOR plays a protective role in mitigating neuronal loss and attenuating epileptogenesis in this model of temporal lobe epilepsy [54] we added the following one: As the authors conclude, the observation that deficiency of mTOR signaling in microglia increases seizure susceptibility contrasts the prevailing opinion that hyperactivation of mTOR is epileptogenic and might explain the inconsistent effect of rapamycin in animal models of epilepsy [63].
Reviewer 4 Report
In my opinion, the review manuscript entitled “Emerging molecular targets for anti-epileptogenic and epilepsy modifying drugs” is very informative as well as generally well organized and well-written. Undoubtedly, the chosen subject of the paper is highly relevant. However, apart from the literature data which were presented in a clear and concise manner, the Authors should also include their own analysis and interpretation. A critical assessment of the presented knowledge, generalization about potential applications, perspectives and directions for future research, and some clear conclusions drawn by the Authors by themselves are needed.
Author Response
Reviewer 4
In my opinion, the review manuscript entitled “Emerging molecular targets for anti-epileptogenic and epilepsy modifying drugs” is very informative as well as generally well organized and well-written. Undoubtedly, the chosen subject of the paper is highly relevant. However, apart from the literature data which were presented in a clear and concise manner, the Authors should also include their own analysis and interpretation. A critical assessment of the presented knowledge, generalization about potential applications, perspectives and directions for future research, and some clear conclusions drawn by the Authors by themselves are needed.
Response: We thank the reviewer for his helpful comments. Generally, we share views of the eminent experts at epilepsy research cited in the manuscript. Nevertheless, the revised version of this manuscript has been supplemented with authors’ personal opinions in the conclusions. It reads: „In our opinion, when looking for antiepileptogenic drug targets, the attention should be paid especially to synaptic and extrasynaptic excitatory and inhibitory amino acid receptors, metabotropic glutamate receptors, adenosine receptor subtypes, neuropeptide receptors, pro-inflammatory cytokines and chemokine receptors. Furthermore, multitargeted drugs which can act at distinct stages of epileptogenesis or rational combinations of single molecular target drugs that prevent pathological excitability and formations of seizure-maintaining neuronal circuits should be considered. Advances in molecular biochemistry, modern electrophysiology, neuroimaging, gene therapy, optogenetic tools and connectomics offer hopes for “rewiring” the pathological seizure-generating or seizure-maintaining neuronal circuits to recover their physiological function”.
Round 2
Reviewer 1 Report
Interesting paper! Acceptable! (Please check the superscripts and subscripts, e.g., Na(+) or IK(erg)).
Author Response
Thank you very much for your kind remarks. The superscripts and subscripts, e.g., Na(+) or IK(erg)) have been corrected in the revised paper.
Reviewer 3 Report
The authors address many of the reviewers concerns, except of re-organizing table 1.
There are references missing (Loscher and Klein) or misplaced. Highly recommend that they should be placed as soon as they are discussed.
Paragraphs should be simplified. For example 2 paragraph introduction (EAetiological factors identifiable in around 50% of epileptic patient...) has too many ideas as to loose the reader of what is the point being made
Author Response
Thank you very much for your insightful suggestions and comments. The table 1 has been re-organized, and the aniseizure medications have been grouped according to their chemical classification, e.g. barbiturates, iminostilbenes, benzodiazepines etc. The information on Aetiological factors identifiable in around 50% of epileptic patient has been shortened. The missing or misplaced references (Loscher and Klein) have been corrected.